

# Comparison of OpenFOAM and EllipSys3D for neutral atmospheric flow over complex terrain

Dalibor Cavar[1], Pierre-Elouan Réthoré[1], Andreas Bechmann[1], Niels N. Sørensen[1], Benjamin Martinez[2], Frederik Zahle[1], Jacob Berg[1], and Mark C. Kelly[1]

[1]Technical University of Denmark, Wind Energy Department, Risø Campus, DK-4000 Roskilde, Denmark
[2]Vattenfall Nordic R&D, DK-7000 Fredericia, Denmark

*Correspondence to:* Dalibor Cavar (daca@dtu.dk)

**Abstract.** The flow solvers OpenFOAM and EllipSys3D are compared in the case of neutral atmospheric flow over terrain using the test cases of Askervein and Bolund hills. Both solvers are run using the steady-state Reynolds-Averaged Navier-Stokes $k$-$\epsilon$ turbulence model.

One of the main modeling differences between the two solvers is the wall-function approach. The OpenFOAM v.1.7.1
uses a Nikuradse's sand roughness model, while EllipSys3D uses a model based on the atmospheric roughness length. It is found that Nikuradse's model introduces an error dependent on the near-wall cell height. To mitigate this error the near-wall cells should be at least 10 times larger than the surface roughness. It is nonetheless possible to obtain very similar results between EllipSys3D and OpenFOAM v.1.7.1. A more recent OpenFOAM v.2.2.1, which includes the atmospheric roughness length wall-function approach, has also been tested and compared to the results of OpenFOAM v.1.7.1 and
EllipSys3D.

The numerical results obtained using the same wall-modeling approach in both EllipSys3D and OpenFOAM v.2.1.1 proved to be almost identical.

Two meshing strategies are investigated, using HypGrid and SnappyHexMesh. The performance of OpenFOAM on SnappyHexMesh based low aspect ratio unstructured meshes is found to be almost an order of magnitude faster than on
HypGrid based structured and high aspect-ratio meshes. However, proper control of boundary layer resolution is found to be very difficult when the SnappyHexMesh tool is utilized for grid generation purposes.

The OpenFOAM is generally found to be $2-6$ times slower than EllipSys3D in achieving numerical results of the same order of accuracy on similar or identical computational meshes, when utilization of EllipSys3D default grid sequencing procedures is included.

**1   Introduction**

Wind resource assessment is of major importance to assess the economic viability of a wind farm project. The traditional tools used by the wind industry rely on linearized flow models and do not perform accurately in complex terrain (Bechmann et al. (2011)). Thus there is a growing interest from the wind industry to use full non-linear Computational Fluid Dynamics (CFD) solutions.



Several CFD solvers are currently available to the industry. Among them OpenFOAM (Ope) has recently received a lot of interest both from the research community and the wind industry. Its popularity seems to be caused by firstly its gratuity and its open source license, which is in strong contrast to the most industry oriented CFD solvers; and secondly, by its good performance[1] e.g. at the Bolund Blind Comparison (Berg et al. (2011); Bechmann et al. (2011)). OpenFOAM therefore directly offers accurate results at no licensing costs.

Two questions are however unanswered: how easy and how fast can OpenFOAM produce accurate results. These two questions are of course relative to other flow solvers but they are relevant and important for the wind industry, as they can represent substantial costs in terms of both necessary hardware and manpower required, due to potentially extensive meshing and computational times. In order to address these questions OpenFOAM is compared with EllipSys3D (Sørensen (1995), Michelsen (1992)). EllipSys3D is an in-house CFD flow solver designed from ground-up for wind energy applications (e.g. atmospheric boundary layer flows, flow over complex terrain and wind turbine rotor computations) at DTU Wind Energy (former Risø National Laboratory (Sørensen (1995)), and DTU-MEK (Michelsen (1992))).

The comparison presented here is focused on the mesh requirements of OpenFOAM relative to EllipSys3D and how this can affect the accuracy of the results. Furthermore, the two CFD codes are benchmarked in speed on the computer cluster at DTU Wind Energy.

In order to keep the comparison as close as possible the two flow solvers are run, to the possible extend, using the same models and parameters. The main model difference between the two flow solvers is the wall-function applied for modeling the effect of the ground roughness on the flow. EllipSys3D (Sørensen et al. (2007a)) uses a wall-function based on the atmospheric roughness length (Richards and Hoxey (1993)) while OpenFOAM v.1.7.1 uses the Nikuradse sand roughness model (Nikuradse (1950)). In OpenFOAM v.2.1.1 a wall-function based on the atmospheric roughness length (Richards and Hoxey (1993)) is also implemented. It should be noted that difference in wall-function modeling has a significant impact on the mesh requirement concerning the height of the first cell above the ground level.

## 2 Methods

### 2.1 Basic equations

Both EllipSys3D and OpenFOAM are based on the finite-volume solution of the Reynolds-averaged Navier-Stokes (RANS) equations. If molecular viscosity is neglected due to the high Reynolds number and the eddy-viscosity hypothesis of Boussinesq (Boussinesq (1877)) is followed, then the RANS equations can be written as,

$$\frac{\partial \overline{u}_i}{\partial t} + \frac{\partial (\overline{u}_i \overline{u}_j)}{\partial x_j} = -\frac{\partial \overline{p}}{\partial x_i} + \frac{\partial}{\partial x_j}(2\nu_T \overline{S}_{ij}) + \overline{f}_i , \qquad (1)$$

where the Einstein summation notation is used. $\overline{u}_i = (\overline{u}, \overline{v}, \overline{w})$ denotes the mean velocity vector, $x_i = (x, y, z)$ are axes of the coordinate system with $z$ being the vertical direction, $\overline{p}$ is the dynamic pressure, $\overline{f}_i$ represents body forces, $\overline{S}_{ij}$ is the

---

[1]Using a non-default, Richards and Hoxey (1993) based, user implemented wall-function formulation.





strain rate tensor and $\nu_T$ is the eddy-viscosity which needs to be modeled. The transient term of Eq. 1 has been retained even though the equations in this work are solved in the steady-state mode.

The classical two-equation high Reynolds number $k - \epsilon$ model (Launder and Spalding (1974)) is utilized in both flow solvers to calculate the eddy-viscosity. Transport equations for the turbulent kinetic energy, $k$, and its dissipation $\epsilon$, used

by $k - \epsilon$ turbulence model are solved and have the following form,

$$\frac{\partial k}{\partial t} + \frac{\partial}{\partial x_j}(\overline{u}_j k) - \frac{\partial}{\partial x_j}\left(\frac{\nu_T}{\sigma_k}\frac{\partial k}{\partial x_j}\right) = \nu_T |\overline{S}|^2 - \epsilon \, , \tag{2}$$

$$\frac{\partial \epsilon}{\partial t} + \frac{\partial}{\partial x_j}(\overline{u}_j \epsilon) - \frac{\partial}{\partial x_j}\left(\frac{\nu_T}{\sigma_\epsilon}\frac{\partial \epsilon}{\partial x_j}\right) = C_{\epsilon 1}\frac{\epsilon}{k}\nu_T |\overline{S}|^2 - C_{\epsilon 2}\frac{\epsilon^2}{k} \, . \tag{3}$$

The originally proposed model constants by Launder and Spalding (1974) were established for industrial flows, while

slightly different values are appropriate for atmospheric flows (Bechmann and Sørensen (2010)). Two test cases are simulated in the present work; the Askervein hill and the Bolund hill. Calibrated model constants are used for the Askervein hill while standard atmospheric values are used for the Bolund hill (see tab. 1 and 4). Identical constants have been used in both EllipSys3D and OpenFOAM based calculations.

### 2.2 Boundary Conditions

In order to model the effect of surface roughness and avoid resolving the laminar sub-layer it is common practice to use the wall-function for atmospheric flows. One of the major differences between EllipSys3D and OpenFOAM v.1.7.1 is related to how the wall-function is implemented. In the more recent OpenFOAM v.2.1.1 the same wall-function as the one in EllipSys3D is included in the official release.

#### 2.2.1 EllipSys3D

In EllipSys3D the traditional high Reynolds number equilibrium assumptions are used to derive the wall-function (see Sørensen (1995); Sørensen et al. (2007b); Hackman (1982) for details). These inherently neglect the laminar sub-layer, which usually is less than a millimeter thick for atmospheric flows. The logarithmic equilibrium profiles used for the mean wind speed, $s$, and turbulent kinetic energy are:

$$s = \frac{u_*}{\kappa}\ln\left(\frac{z + z_0}{z_0}\right) \, , \tag{4}$$

$$k = \frac{u_*^2}{C_\mu^{1/2}} \, , \tag{5}$$

where $z_0$ is the roughness length, $\kappa = 0.40$ is the von Karman constant and $u_*$ is the friction velocity.

As seen from Eq. 4, the wall is placed on top of the roughness elements ($s = 0$ for $z = 0$) and is consequently displaced by the roughness length. This has been done to avoid a minimum height restriction of the first computational cell (see





Section 2.2.2) and EllipSys3D can thereby resolve large near-wall velocity gradients using shallow (high aspect ratio) computational cells.

In order not to abandon the momentum equation in the near-wall cell by simply fixing the velocity according to log-law (Eq. 4), EllipSys3D instead implements the wall-function through the wall shear stress, $\tau_0$. Based on Eq. 4 and 5 the

kinematic shear stress is calculated from $\tau_0 = \rho\, u_0\, u_w$, using,

$$u_0 = k^{\frac{1}{2}} C_\mu^{\frac{1}{4}} \, , \tag{6}$$

$$u_w = \frac{\kappa s}{\ln\left(\frac{\Delta z + z_0}{z_0}\right)} \, , \tag{7}$$

where $\Delta z$ is the distance from the bottom cell face to the cell center. The $\epsilon$-equation is abandoned in the near-wall cells;

instead $\epsilon$ is specified according to,

$$\epsilon = \frac{u_0^3}{\kappa\left(\Delta z + z_0\right)} \, , \tag{8}$$

while the $k$-equation is reduced to a balance between production ($P_k = \nu_T |\overline{S}|^2$) and dissipation $\epsilon$ and a von Neumann boundary condition is set for $k$. The boundary condition for the velocity is the standard no-slip condition ensuring the velocity to be zero at the wall.

**2.2.2 OpenFOAM**

### 2.2.3 OpenFOAM v.1.7.1 - Nikuradse's equivalent sand roughness length model

The wall-function available in OpenFOAM v.1.7.1 is based on Nikuradse's equivalent sand roughness length, $k_s$, (Nikuradse (1950)) and can be used to model flows where the laminar sub-layer is important. The OpenFOAM implementation of an atmospheric equilibrium boundary-layer condition, according to the expression for surface roughness of Nikuradse,

reduces to the following equation for the mean wind speed[2] (Martinez (2011); Blocken et al. (2007)),

$$s = \frac{u_*}{\kappa} \ln\left(\frac{Ez}{C_s k_s}\right) \approx \frac{u_*}{\kappa} \ln\left(\frac{z}{z_0}\right) \, , \tag{9}$$

where $C_s = 0.5$ is a roughness constant, $E = 9.79$ is a smooth wall constant and $k_s = \frac{E}{C_s} z_0 = 19.58\, z_0$ have been used.

Comparing with (Eq. 4) it is seen that the roughness for OpenFOAM v.1.7.1 is placed on top of the wall ($s = 0$ for $z = z_0$). While this is physically the 'correct' approach it does set some constraints on the heights of the near-wall cells. Firstly, as

argued in Blocken et al. (2007), it is not physically meaningful to have grid cells within the roughness height, therefore the height of the near-wall cell center should be at least equal or larger than the Nikuradse's roughness length ($\Delta z \geq k_s$). Blocken et al. (2007) does however state, that it is mathematically/numerically possible to have $\Delta z \leq k_s$ and comparison

---

[2]Assuming $C_s k_s^+ \gg 1$, with $k_s^+ = \frac{u_* k_s}{\nu}$ and $\nu$ is a molecular viscosity





between measurements and a sand grain based velocity function fit - (Sumer (2007)), shows that their agreement is very good when $\Delta z \geq 0.2\,k_s$. Secondly, OpenFOAM has a restriction on the cell aspect ratio, i.e. the ratio between the longest and the shortest cell dimension. Meshes with cell aspect ratio larger than 1000 have a tendency to introduce numerical errors and to converge slower (Martinez (2011)) and it is therefore difficult to make shallow cells that resolve the flow close

to the ground. As a compromise between the cell constraints and the need to resolve the near-wall flow, $\Delta z/z_0 \approx 10.0$, was found as rule of thumb to give the least numerical error for flat terrain simulations (Martinez (2011)) and is used in the following.

Using the build-in routine (`nutRoughWallFunction`), OpenFOAM v.1.7.1 implements the wall-function by specifying the eddy-viscosity that for fully rough flows can be written as,

$$\nu_T = \frac{u_* \kappa \Delta z}{\ln\left(\frac{E\Delta z}{C_s k_s}\right)} \approx \frac{u_0 \kappa \Delta z}{\ln\left(\frac{\Delta z}{z_0}\right)} \,. \tag{10}$$

Similar to EllipSys3D, the $\epsilon$-equation is abandoned in the near-wall cells. Using the build-in function (`epsilonWallFunction`), $\epsilon$ is set to,

$$\epsilon = \frac{u_0^3}{\kappa \Delta z} \,, \tag{11}$$

while a zero gradient condition is set on $k$ (using the `kqRWallFunction`). A no-slip condition is set on the velocity

vector.

### 2.2.4   OpenFOAM v.2.1.1 - Richards and Hoxey (1993) based model

The model is identical to the model implemented in EllipSys3D. Regarding the OpenFOAM implementation, the only difference compared to Nikuradse's roughness length model is in the way turbulent viscosity is determined.

Using the build-in routine (`nutkAtmRoughWallFunction`), OpenFOAM v.2.1.1 implements the wall-function by

specifying the eddy-viscosity as

$$\nu_T = \frac{u_0 \kappa \Delta z}{\ln\left(\frac{\Delta z + z_0}{z_0}\right)} \,. \tag{12}$$

All other boundary conditions used in connection with this wall-model are identical to the ones described in the previous paragraph.

### 2.3   Solution techniques

In the simulations carried out in this work, in both flow solvers, the RANS equations are discretized using the QUICK scheme (Leonard (1979)) (`QUICKV`, the vectorial version for OpenFOAM). The $k$ and $\epsilon$ equations are discretized using the first order Upwind Discretization Scheme (UDS). The pressure is solved using the SIMPLE algorithm (Patankar and Spalding (1972)) and is accelerated using a multigrid approach (`GAMG` for OpenFOAM).





| $u_*$ [m/s] | $z_0$ [m] | $\kappa$ | $C_\mu$ | $\sigma_k$ | $\sigma_\varepsilon$ | $C_{\varepsilon 1}$ | $C_{\varepsilon 2}$ |
|---|---|---|---|---|---|---|---|
| 0.6110 | 0.03 | 0.4 | 0.119 | 1.000 | 1.301 | 1.564 | 1.920 |

**Table 1.** Set of atmospheric parameters used for Askervein, as described in Taylor and Teunissen (1987)

### 2.4 Mesh Generators

#### 2.4.1 HypGrid

EllipSys3D uses the mesh generator HypGrid (Sørensen (1998)), a hyperbolic mesh generator developed at DTU Wind Energy. HypGrid creates a 3D structured hexahedron volume mesh using a hyperbolic marching scheme, based on orthogonality and cell volume from a 3D terrain grid surface definition. It can produce meshes with cells of low non-orthogonality and low skewness. There is no constraint on cell aspect ratio.

#### 2.4.2 SnappyHexMesh

SnappyHexMesh is an hexahedron unstructured mesh generation tool included in the distribution of OpenFOAM. Snap-pyHexMesh can use a 3D terrain surface and iteratively build a mesh upon it. Some options allow construction of several cell layers of a controllable height above the terrain surface. This feature makes it possible to have a refined mesh in the region of high velocity gradients, close to the ground. Several regions can be selected to be refined to a desired level, and this method is creating a mesh of a cell aspect ratio close to one. This is done in order to ensure that OpenFOAM can solve the numerical problems with the highest efficiency and accuracy using meshes generated with SnappyHexMesh. The meshing tool SnappyHexMesh can be run in parallel on a computer cluster or a PC.

### 3 Results

#### 3.1 Askervein

#### 3.1.1 Simulation inputs

The inputs used to simulate the Askervein hill are fitted based on the upstream mast measurements.

The complete set of all simulation inputs, in accordance with Taylor and Teunissen (1987), is found in tab. 1.




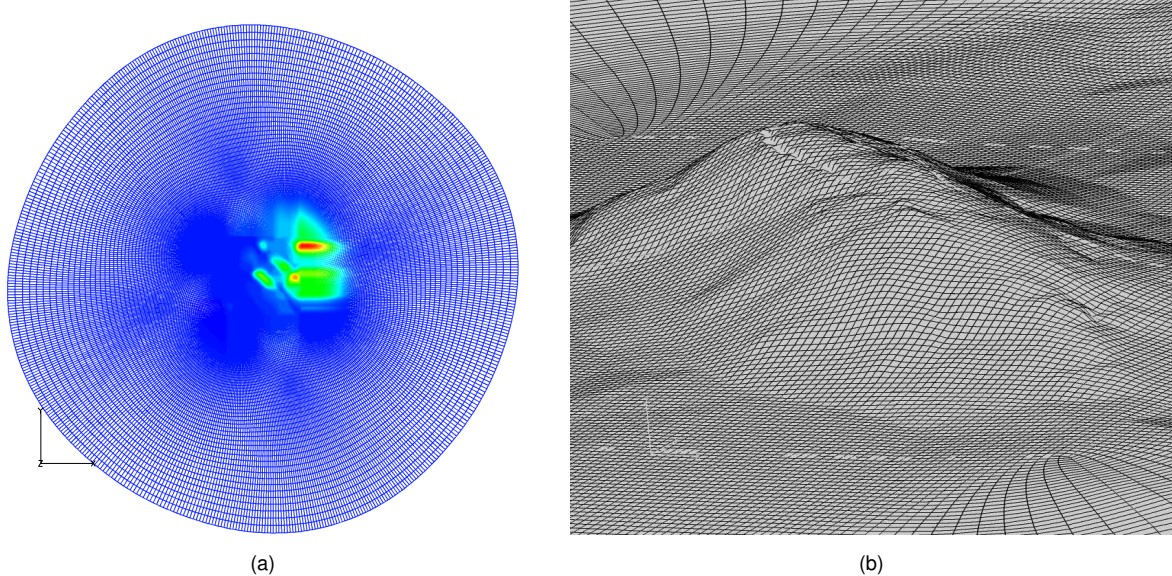

(a)                                                                (b)

**Figure 1.** Askervein hill: Surface grid generated with surfgrid. a) general view of the grid b) close look-up of the grid in the hill

### 3.1.2 Description of the mesh

### 3.1.3 HypGrid

The Askervein map is used as input to the in-house tool `surfgrid`. The `surfgrid` generated a circular (32.5 km in diameter) surface mesh, with a refined region of 2×2 km (4 blocks) in the center, at the position where the Askervein hill

is located (see fig. 1). From this surface mesh the *HypGrid* tool is used to hyperbolically march the surface into a 6.5 km high 3D mesh. The final 3D volume mesh is composed of 20 blocks of 64x64x64: 5.2M cells (see fig. 2).

     Two meshes are created with this method, one with a first cell center height of $0.4\,\mathrm{m} = 13.3\,z_0$ (ideal for OpenFOAM v.1.7.1 - the HG1 grid), and one with a first cell center height equal to a half of the roughness length $z_0 = 0.03$ m (ideal for EllipSys3D - the HG2 grid). Two additional meshes (with the first cell center height of $\Delta z = 0.83\,z_0$ and $\Delta z = 1.5\,z_0$) were

created for OpenFOAM v.2.1.1, but they both failed several grid compliance tests performed by `checkMesh` OpenFOAM tool. A converged solution could however be obtained on those grids only if the `mapFields` tool is used to e.g. interpolate final solution obtained on the grid with the first cell center height at $\Delta z = 13.3\,z_0$. As the proper speed convergence tests could not be conducted on the mentioned grids, the numerical results based on them are not included here. The coarse HG0 grid (tab. 2) is created exclusively to accommodate a proper grid sequencing procedure in the OpenFOAM HypGrid

based computations.





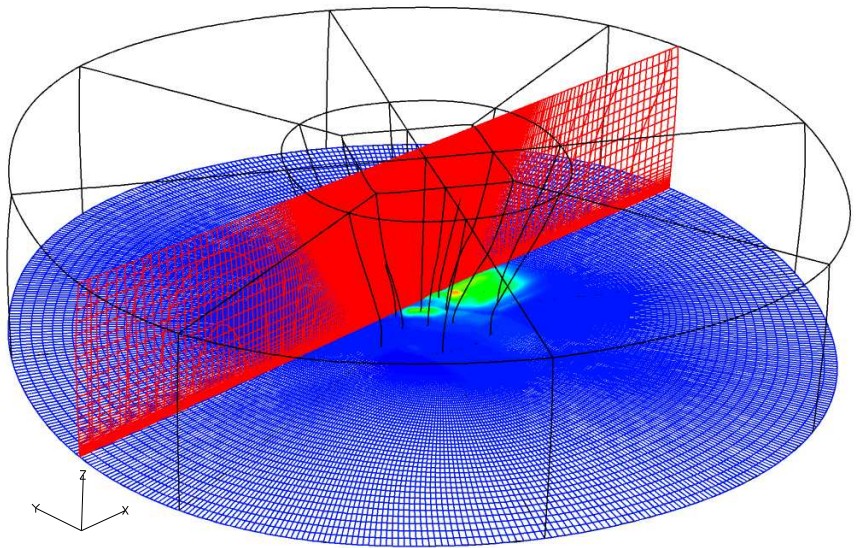

**Figure 2.** Askervein hill: Volume grid generated with HypGrid

### 3.1.4 SnappyHexMesh

The `surfgrid` generated surface grid used for HypGrid meshing is converted to STL file format and utilized to generate a ground boundary surface suitable for the SnappyHexMesh OpenFOAM utility. A rectangular domain of $11.03 \times 11.03 \times 3.10\,\mathrm{km}$ is discretized using the `blockMesh` utility (see tab. 2) creating a background mesh with resolution of $95.9 \times 95.9 \times 38.75\,\mathrm{m}$

in $x$, $y$, $z$ directions, respectively. Only the SHM4 mesh (tab. 2) has a background mesh resolution of $77.5\,\mathrm{m}$ in $z$ direction. The cross-section of the SnappyHexMesh created grids with indicators locating positions of refinement boxes and surface layers used to generate meshes SHM(0-4) is presented in fig. 3. Changing refinement levels in refinement boxes 1-3 together with the number of inserted surface layers are the controlling parameters used in the SnappyHexMesh grid creation process (tab. 2). The `refinementSurface` and `resolveFeatureAngle` parameters are used to control

the surface refinement level relative to the background grid. The coarse SHM0 (tab. 2) grid is only intended for use in connection with grid sequencing procedures.

As it can be seen from tab. 2, grids with optimal position of the first near ground cell heights, suitable for both Open-FOAM v.1.7.1 and v.2.1.1, could be created in this way.

### 3.1.5 Simulation results

The main results of the Askervein test case are presented in fig. 4 and fig. 5. Both figures show the results of the simulations compared with the cup anemometer measurements along the line A - see Taylor and Teunissen (1987). The



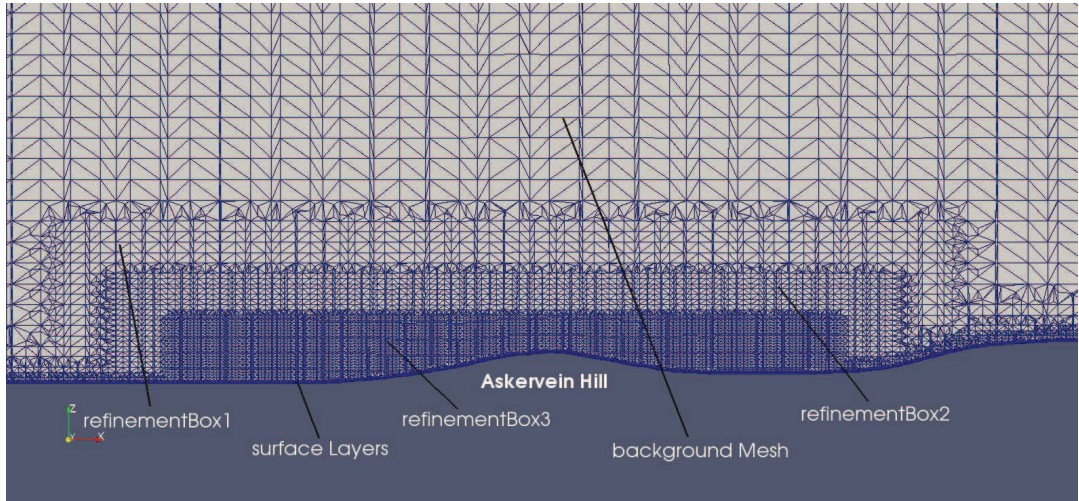

**Figure 3.** Askervein hill: Cross section of the grid created by SnappyHexMesh. Three refinement boxes together with variations in number of surface layers indicated in the figure are used as a basis for generation of the grids described in tab. 2

basic simulations are carried out on HypGrid based grids HG(1-2) and SnappyHexMesh based grids SHM(1-4). On fig. 4a) and fig. 5a) EllipSys3D calculation on structured mesh HG1 and OpenFOAM (both v.1.7.1 and v.2.1.1) computations on the same mesh are presented. The lines in red, black and blue colors represent therefore the results of OpenFOAM and EllipSys3D calculations using the identical mesh.

Overall, the results of all simulations are very similar to the measurements in the region upstream the hill-top, both in terms of speed-up and turbulent kinetic energy (TKE).

After the hill-top however, results of the simulations start to differ from each other significantly, in terms of speed-up and to a lesser extend in terms of TKE. In comparison with the measurements, the speedup results are relatively close to the EllipSys3D simulation, using the mesh with a first cell height equal to the surface roughness (HG2 grid - dotted magenta

line - fig. 4a)) and OpenFOAM v.2.1.1 SnappyHexMesh SHM(2-4) based simulations - fig. 4b). Both OpenFOAM (v.1.7.1 and v.2.1.1) and EllipSys3D computations based on HG1 grid have the largest deviations from the measurements in this area. In terms of TKE, all simulations are about half the value obtained by the measurements.

One should also note a very good agreement between OpenFOAM v.2.1.1 and EllipSys3D (the dashed black and blue curves), based on identical wall-function (Richards and Hoxey (1993)) model and calculated on identical computational

meshes (HG1), especially in terms of speed-up (fig. 4a)) but also in terms of TKE (fig. 5a)).

### 3.1.6   Simulation time

Default values, (i.e. from `simpleFoam` tutorials) regarding basic OpenFOAM solver inputs are used in all OpenFOAM based calculations. Several attempts have been made to change/tweek some of (many) multigrid pressure solver - `GAMG` parameters, but they all resulted in prolonged computational times and in some cases led to a periodic rather than





| | block Mesh (back-ground grid) | refine-ment Surfa-ces | re-solve Fea-ture Angle (°) | ref. Box 1 | ref. Box 2 | ref. Box 3 | Grid Size Prior Add Surf. Layer (mill.) | nSu-rface La-yers | Total Height of Grid Layer (m) | Grid Size Add Layer (Total) (mill.) | $\Delta z$ / $z_0$ |
|---|---|---|---|---|---|---|---|---|---|---|---|
| SHM0 | 60,60,40 | level (2 2) | 3 | 0 | 0 | 0 | 0.31 | 6 | 7.98 | 0.33 (0.64) | 13.3 |
| SHM1 | 115,115,80 | level (2 3) | 3 | 1 | 0 | 2 | 1.69 | 6 | 7.98 | 1.75 (3.44) | 13.3 |
| SHM2 | 115,115,80 | level (2 3) | 3 | 1 | 0 | 2 | 1.69 | 8 | 0.99 | 2.32 (4.01) | 1.0 |
| SHM3 | 115,115,80 | level (2 3) | 3 | 1 | 0 | 3 | 2.10 | 10 | 9.03 | 2.90 (5.00) | 8.3 |
| SHM4 | 115,115,40 | level (3 3) | 3 | 1 | 2 | 3 | 2.53 | 6 | 12.0 | 5.02 (7.55) | 20.0 |
| HG0 | - | - | - | - | - | - | - | - | - | (1.31) | 13.3 |
| HG1 | - | - | - | - | - | - | - | - | - | (5.24) | 13.3 |
| HG2 | - | - | - | - | - | - | - | - | - | (5.24) | 0.5 |

**Table 2.** Askervein hill: Overview of different control parameters used to generate the SnappyHexMesh created grids. BlockMesh column shows nr. of grid points in $x$, $y$, $z$ directions in the background mesh, respectively. RefinementSurfaces column shows the minimum level of surface refinement relative to the blockMesh created background grid (first number in the parentheses) and the maximum surface refinement level, which is used if a cell intersection angle (angle between two adjacent cells) > resolveFeatureAngle (nr. shown in column three) - second number in the parentheses. RefinementBox 1-3 columns show the local grid refinement level relative to the background grid. Following three columns show grid size before addSurfaceLayer option is used, number of added surface layers and a total height of grid zone corresponding to added surface layers. Last two columns show number of grid points in added surface layers (total number of grid points in the entire mesh shown in the parentheses) and the ratio between the first cell center height and the roughness length. Definition of all grids, including the HypGrid based ones, together with the corresponding grid sizes is also included. Positions of refinement boxes, surface layers and background grid are indicated in fig. 3. The SHM are SnappyHexMesh based grids and HG are HypGrid based ones.

monotonic decay residual behavior. For extensive details about all input parameters considered, the interested reader is referred to Martinez (2011).

The computational process in EllipSys3D has been done both using the standard 5 level grid sequencing[3] and without it. In the OpenFOAM case the direct grid sequencing procedure does not exist. Here, two grids - the first consisting of 32x32x64: 1.3M cells - HG0 grid (tab. 2) has been used in connection with OpenFOAM calculations on HG1 grid and the second one - SHM0 (tab. 2) is used in connection with SnappyHexMesh based OpenFOAM calculations. Upon reaching a suitable convergence level on HG0 and SH0 grids, the OpenFOAM's `mapFields` tool has then been used to interpolate the results to the fine HG(1-2) and SHM(1-4) grids. The computational process on the fine grids is then continued until

[3]A built-in EllipSys3D function. It can easily be used to directly perform e.g. grid dependency study.





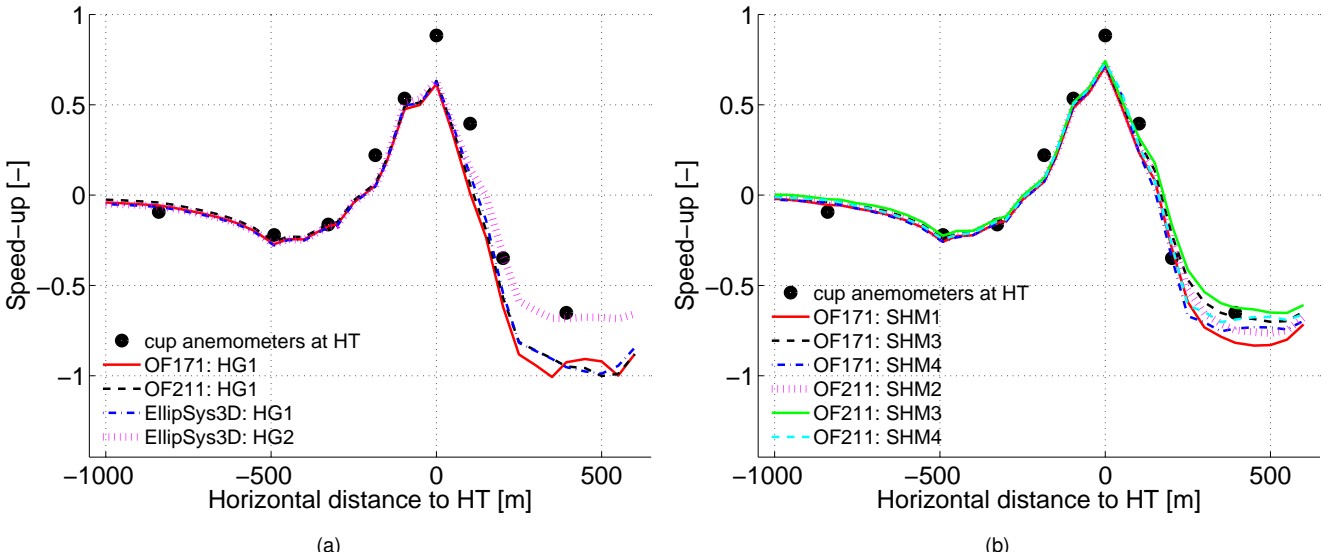

**Figure 4.** Askervein Hill: Speed-up along the line A. For legend see fig. 5.

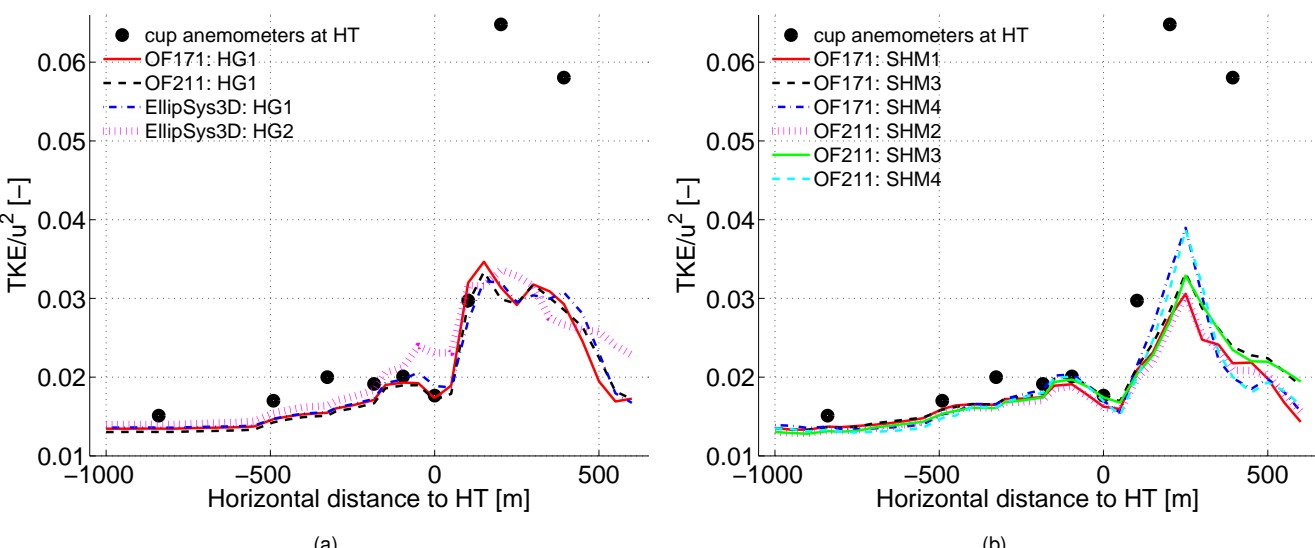

**Figure 5.** Askervein Hill: Turbulent kinetic energy along the line A.

the convergence criterion is met. Also, the OpenFOAM computations are carried out without using the aforementioned procedure i.e. solving on the fine grid using the standard initial values for all variables.

The obtained computational times are presented in tab. 3.





| | | | Grid sequencing ON | Grid sequencing OFF | Grid Size (mill.) |
|---|---|---|---|---|---|
| EllipSys3D : | HG2 | $\Delta z = 0.50\,z_0$ | 509 s | 826 s | 5.24 |
| EllipSys3D : | HG1 | $\Delta z = 13.3\,z_0$ | 454 s | 754 s | 5.24 |
| OF v.1.7.1 : | HG0 | $\Delta z = 26.6\,z_0$ | - | 1323 s | 1.31 |
| OF v.1.7.1 : | HG1 | $\Delta z = 13.3\,z_0$ | 6655 s | 10259 s | 5.24 |
| OF v.1.7.1 : | SHM0 | $\Delta z = 13.3\,z_0$ | - | 68 s | 0.64 |
| OF v.1.7.1 : | SHM1 | $\Delta z = 13.3\,z_0$ | 693 s | 1167 s | 3.44 |
| OF v.1.7.1 : | SHM3 | $\Delta z = 8.33\,z_0$ | 989 s | 1909 s | 5.00 |
| OF v.1.7.1 : | SHM4 | $\Delta z = 20.0\,z_0$ | 1867 s | 4311 s | 7.55 |
| OF v.2.1.1 : | HG1 | $\Delta z = 13.3\,z_0$ | 3042 s | 14421 s | 5.24 |
| OF v.2.1.1 : | SHM2 | $\Delta z = 1.00\,z_0$ | 527 s | 1037 s | 4.01 |
| OF v.2.1.1 : | SHM3 | $\Delta z = 8.33\,z_0$ | 862 s | 2013 s | 5.00 |
| OF v.2.1.1 : | SHM4 | $\Delta z = 20.0\,z_0$ | 1591 s | 3447 s | 7.55 |

**Table 3.** Askervein hill: Simulation times for EllipSys3D and OpenFOAM codes. For information about different grids utilized in the computations see tab. 2 and fig. 3.

| $u_* \, [m/s]$ | $z_0 \, [\text{m}]$ | $\kappa$ | $C_\mu$ | $\sigma_k$ | $\sigma_\varepsilon$ | $C_{\varepsilon1}$ | $C_{\varepsilon2}$ |
|---|---|---|---|---|---|---|---|
| | $3\,\text{x}\,10^{-4}$ for $z < 0.8\,\text{m}$ | | | | | | |
| 0.4 | $1.5\,\text{x}\,10^{-2}$ for $z \geq 0.8\,\text{m}$ | 0.4 | 0.03 | 1.0 | 1.30 | 1.21 | 1.92 |

**Table 4.** Set of atmospheric parameters used for Bolund hill test case.

Comparing EllipSys3D to OpenFOAM runs (tab. 3) especially the fastest ones obtained on grids of similar size and location of the first near ground computational cell - EllipSys3D HG1 grid and OpenFOAM SHM3 grid, it can be observed that EllipSys3D is app. factor $1.9 - 2.5$ times faster in obtaining the numerical solution of the same level of accuracy.

### 3.2   Bolund

5   **3.2.1   Simulation inputs**

The inputs used to simulate Bolund hill test case are based on quantities proposed by Bechmann et al. (2011). The values used are presented in tab. 4.



### 3.2.2 Description of the mesh

### 3.2.3 HypGrid

A Bolund map used in connection with the Blind Comparison Test presented in Bechmann et al. (2011), is used as a basis for the generation of the surface mesh. The `surfgrid` tool could be readily used to generate an appropriate surface
mesh for the EllipSys3D flow solver, but it turned out to be difficult to produce a surface mesh, which could both be used in the EllipSys3D and the OpenFOAM solvers. A high complexity and very abrupt change in the surface structure on the Bolund hill front side caused severe difficulties with regard to obtaining a suitable OpenFOAM 3D meshes, without problems in cell face orientation. The grid validity check conducted using the `checkMesh` OpenFOAM tool showed that bad cells could always be located in the mentioned area. As `surfgrid` tool does not have an option to visualize the
grid during the creation process nor provide a possibility to smooth the created grids, the Pointwise's *GridGen* mesh generation software has thus been used for the purpose of smoothing the `surfgrid` generated mesh[4]. Depiction of the whole surface grid and a close-up view at the Bolund island are presented on fig. 6. The 4 km in diameter surface mesh - fig 6a), is centered on and refined around the Bolund island position. The *HypGrid* tool is used to hyperbolically march the grid in the third dimension (up to a height of 1 km). The final volume grid - fig. 7 comprise of 24 blocks of 64
x 64 x 64 cells, app. 6.3M grid points. Basically three meshes are created in this way, one with the first cell center height of $\Delta z = 0.1875\,\mathrm{m}$ i.e. $k_s(z_{0(0.0003)}) \approx 0.006\,\mathrm{m} < \Delta z < k_s(z_{0(0.015)}) = 0.294$ (OpenFOAM v.1.7.1 - the HG1 grid), second one with the first cell center height of $\Delta z = 0.0125\,\mathrm{m}$ i.e. $z_{0(0.0003)} < \Delta z < z_{0(0.015)}$ (OpenFOAM v.2.1.1 - the HG2 grid) and the third one with the first cell center height of $\Delta z = 0.0005\,\mathrm{m}$ i.e. $z_{0(0.0003)} < \Delta z < z_{0(0.015)}$ (EllipSys3D - the HG3 grid).
As it can be seen, some compromises on the position of the first grid point in the surface normal direction (especially in the OpeanFOAM cases) had to be made due to the change in the surface roughness length throughout the computational domain. Even though no formal model based restriction regarding the position of the first cell center height in the Richards and Hoxey (1993) implementation of the wall-function in OpenFOAM v.2.1.1 exist, the previously discussed aspect ratio issue dictated that $\Delta z = 0.0125\,\mathrm{m}$. With this $\Delta z$ value the max aspect ratio is app. 7000, and the only error/warning
reported by `checkMesh` tool was regarding the cell aspect ratio. Diminishing the $\Delta z$ value further, introduces several other mesh related errors reported by the `checkMesh` tool and any attempt to obtain the numerical solution resulted in almost immediate divergence.

### 3.2.4 SnappyHexMesh

A similar procedure as in the Askervein hill test case, regarding the re-use of the `surfgrid` created ground surface
mesh utilized in the HypGrid meshing tool, is conducted in the Bolund hill case. Upon the surface grid conversion to STL file format, a rectangular domain of $2.3\,\mathrm{x}\,2.3\,\mathrm{x}\,1.0\,\mathrm{km}$ is discretized using the `blockMesh` utility (see tab. 5) creating a

---

[4]It turned out that very limited number of cells very close to the Bolund hill front edge needed to be rearranged. This was done by still keeping the fixed surface projection, ensuring that the created grid had the same qualitative Bolund hill surface representation.




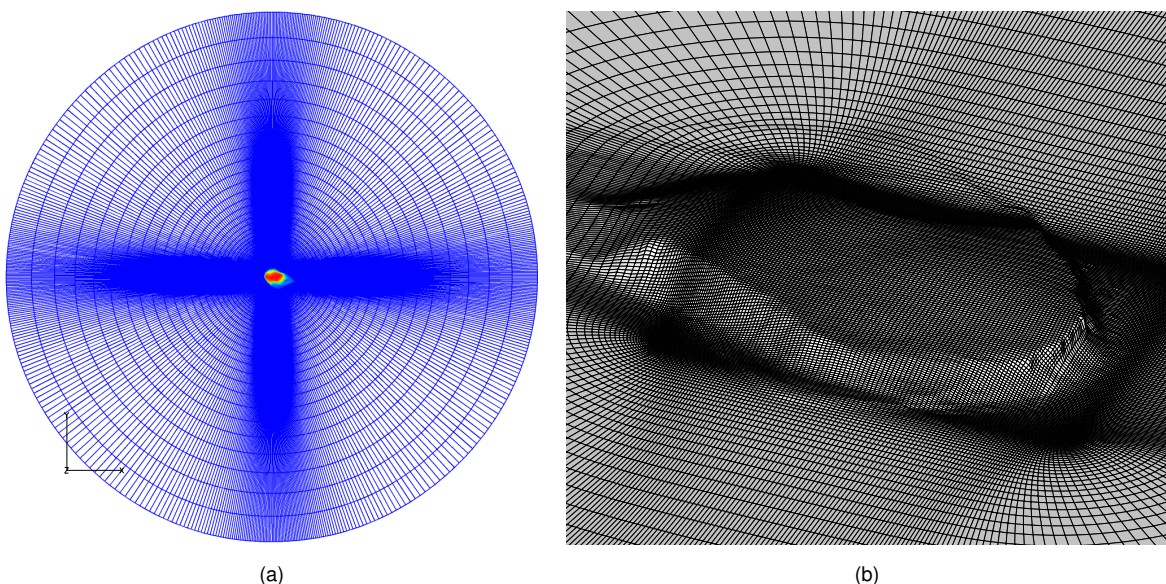

**Figure 6.** Bolund hill: Surface grid generated by the *HypGrid* and smoothed by the *GridGen*. a) general view of the grid b) close look-up of the grid in the hill

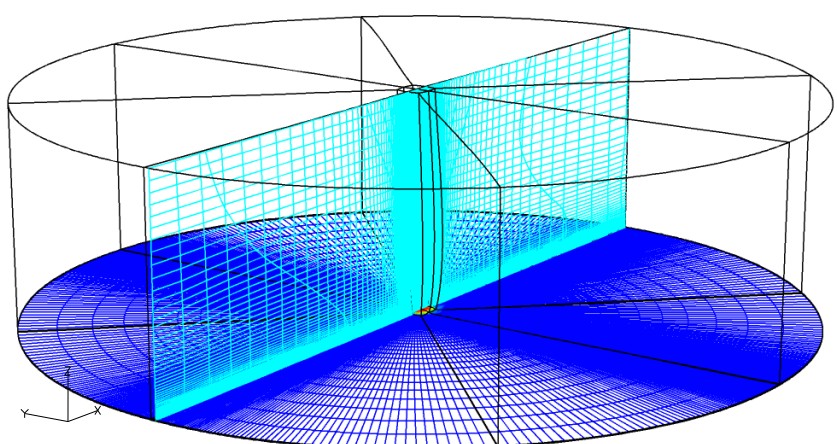

**Figure 7.** Bolund hill: Volume grid generated with the *HypGrid*

background mesh with resolution of $30.7\,\mathrm{x}\,30.7\,\mathrm{x}\,16.7\,\mathrm{m}$ in $x$, $y$, $z$ directions, respectively. Analogously to the Askervein hill case, the cross-section of the SnappyHexMesh created grids with indicators locating positions of refinement boxes and surface layers used to generate meshes SHM(0-3) are presented in fig. 8. Similarly, changing the refinement levels in refinement boxes 1-3 together with number of inserted surface layers are the controlling parameters used for the Snap-





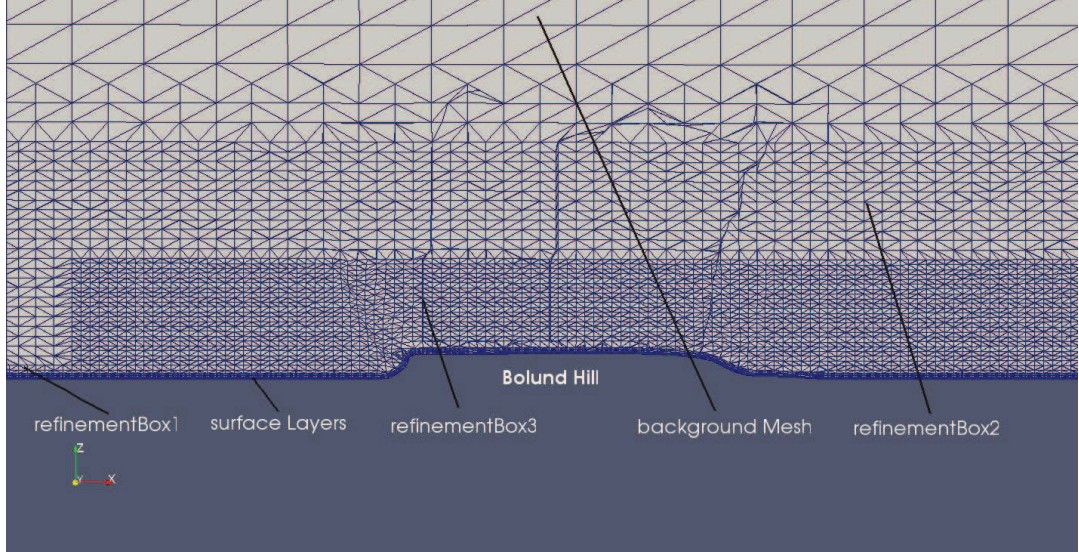

**Figure 8.** Bolund hill: Cross section of the grid created by SnappyHexMesh. Three refinement boxes together with variations in number of surface layers indicated in the figure are used as a basis for generation of the grids described in tab. 5

pyHexMesh grid creation (tab. 5). The `refinementSurface` and `resolveFeatureAngle` parameters are used to control the surface refinement level relative to the background grid. They seem to play a much more important role in the Bolund hill case than previously, apparently due to the sudden and abrupt change in the surface topology characterizing the Bolund hill case. It should be noted that increasing the surface refinement level to more than level $5$ (tab. 5) led directly to an inability of SnappyHexMesh to create valid surface layers. The coarse SHM0 and HG0 (tab. 5) grids are again only intended for use in connection with grid sequencing procedures.

### 3.2.5 Simulation results

The results of the Bolund test case are presented on fig. 9 and 10. The computations are conducted for the incoming wind flow direction of $270^o$, and results are compared with measurements along the line B (for details see Bechmann et al. (2011)).

The obtained results are compared to the results submitted by other participants of the Bolund Blind Test Contest (not shown here) and found to be in a close agreement with the submitted numerical results attained utilizing two equation turbulence models.

As a roughness change characterizes the Bolund hill test case, some considerations regarding the position of the first grid point in the surface normal direction do exist. The wall function closure used in EllipSys3D flow solver does not restrict the position of the first grid point in the surface normal direction, so the position chosen here corresponds to 3 x roughness length of the water ($z_0 = 0.0003\,\text{m}$) and $1/15$ x roughness length of the land ($z_0 = 0.015\,\text{m}$). In the OpenFOAM



| | block Mesh (background grid) | refinement Surfaces | resolve Feature Angle (°) | ref. Box 1 | ref. Box 2 | ref. Box 3 | Grid Size Prior Add Surf. Layer (mill.) | nSurface Layers | Total Height of Grid Layer (m) | Grid Size Add Layer (Total) (mill.) | $\Delta z / z_{0(L)}$ |
|---|---|---|---|---|---|---|---|---|---|---|---|
| SHM0 | 45,45,50 | level (3 5) | 2 | 0 | 0 | 0 | 0.43 | 3 | 1.60 | 0.41 (0.84) | 13.3 |
| SHM1 | 75,75,60 | level (2 3) | 2 | 3 | 2 | 3 | 3.47 | 6 | 2.61 | 2.13 (5.60) | 10.0 |
| SHM2 | 75,75,60 | level (3 5) | 2 | 0 | 2 | 3 | 3.53 | 3 | 1.60 | 1.15 (4.68) | 13.3 |
| SHM3 | 75,75,60 | level (3 5) | 2 | 0 | 2 | 3 | 3.53 | 11 | 1.74 | 4.10 (7.63) | 1.0 |
| HG0 | - | - | - | - | - | - | - | - | - | (0.78) | 25.0 |
| HG1 | - | - | - | - | - | - | - | - | - | (6.29) | 12.5 |
| HG2 | - | - | - | - | - | - | - | - | - | (6.29) | 0.83 |
| HG3 | - | - | - | - | - | - | - | - | - | (6.29) | 0.03 |

**Table 5.** Bolund hill: Overview of different control parameters used to generate the SnappyHexMesh created grids. For definition of different parameters in the table - see tab. 2. Positions of refinement boxes, surface layers and background grid is indicated in fig. 8. The SHM are SnappyHexMesh based grids and HG are HypGrid based ones. $z_{0(L)} = z_{0-Land} = 0.015\,\mathrm{m}$.

v.1.7.1 case, based on investigations of Martinez (2011), the largest roughness length for land had to be used as a basis for grid creation process in order to avoid problems with the limitations of the Nikuradse's sand roughness length closure ($\Delta z/z_{0-Land} = 12.5$). This however places the first grid point at the inlet and whole region upstream the Bolund island position at a very large relative distance from the terrain ($\Delta z/z_{0-Water} = 625$). This issue could potentially negatively

influence the results upstream the Bolund island position.

For this reason two different 2D flat terrain computations involving the grids where the position of the first grid point in the surface normal direction was varied from $\Delta z/z_{0-Water} = 12.5$ to $\Delta z/z_{0-Water} = 625$ were conducted. The obtained results (not shown here) however, appeared to be in a close agreement with each other. Also comparison of OpenFOAM (both v.1.7.1 and v.2.1.1) and EllipSys3D results at position of mast 7 (M7) - position just upstream the Bolund island

- shown in fig. 11, indicate that the mentioned issue does not seem to have any negative influence on the most of the presented OpenFOAM (both v.1.7.1 and v.2.1.1) results. As seen from fig. 11, only OpenFOAM (both v.1.7.1 and v.2.1.1) results based on SHM1 grid deviate in terms of TKE from the rest of the computations. This issue will be discussed further in the next section.

From figs. 9 and 10 a general good agreement between results of both OpenFOAM v.2.1.1, v. 1.7.1 and EllipSys3D

together with their relative good correspondence with the measurements, can be observed. Largest differences between results can be observed in TKE plots at 2 m above the ground level. This subject will also be discussed further in section



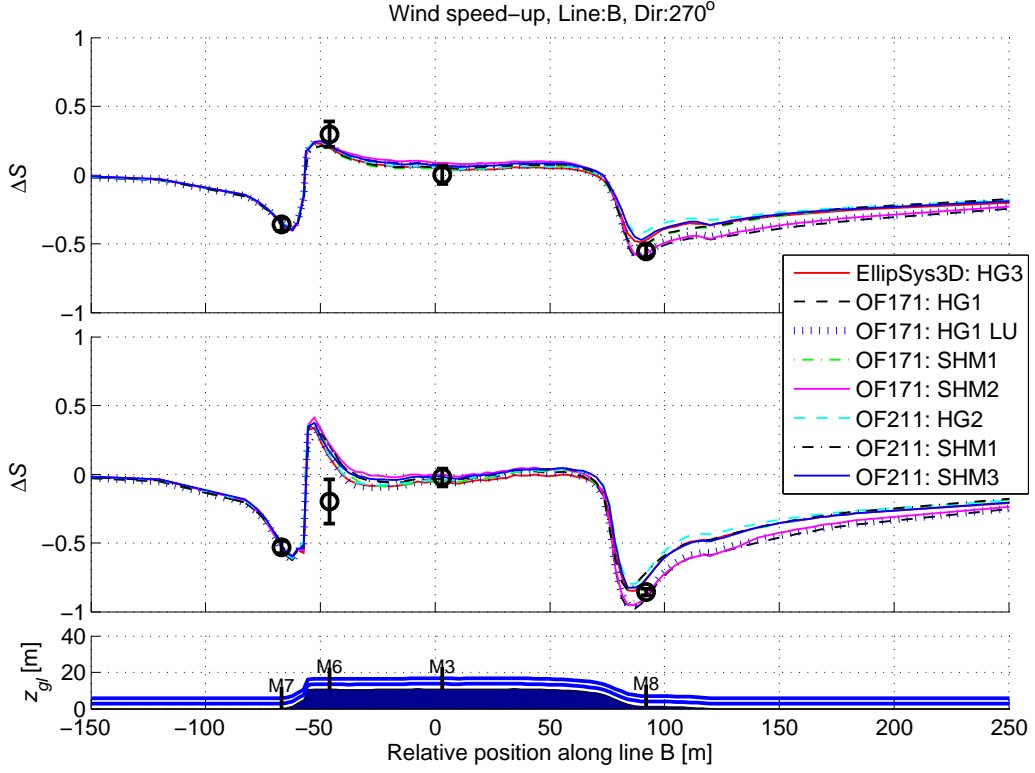

**Figure 9.** Bolund hill: Speed-up along the line B. Uppermost subfigure $5\,\mathrm{m}$ above the ground level, middle subfigure $2\,\mathrm{m}$ above the ground level. Measurements are denoted as circles with corresponding uncertainties denoted as error bar.

4.2. Furthermore, especially results using OpenFOAM v.2.1.1 (HG2 grid) and EllipSys3D, based on identical wall-function model, have a very good agreement. It is also seen that all calculations have similar difficulties in agreements with the measurements in the area immediately after the sharp Bolund hill front edge, as previously reported in Bechmann et al. (2011).

### 3.2.6 Simulation time

The solver inputs, apart from the Bolund hill specific ones presented in tab. 4, regarding both OpenFOAM and EllipSys3D are kept identical to the inputs previously used in the Askervein test case. It should be noted that stable convergence could not be obtained on SnappyHexMesh based grids using the `QUICKV` discretization scheme. For that reason a formally second order `linearUpwindV`[5] scheme was used in those cases. Computations on HypGrid based HG1 grid in OpenFOAM were redone using the `linearUpwindV` scheme in order to be able to access the speed of convergence and quality of the obtained results in a proper manner. The OpenFOAM v.2.2.1 runs on HG2 grid are conducted using the

[5]Specifying: div(phi,U) Gauss linearUpwindV cellMDLimited Gauss linear 1;





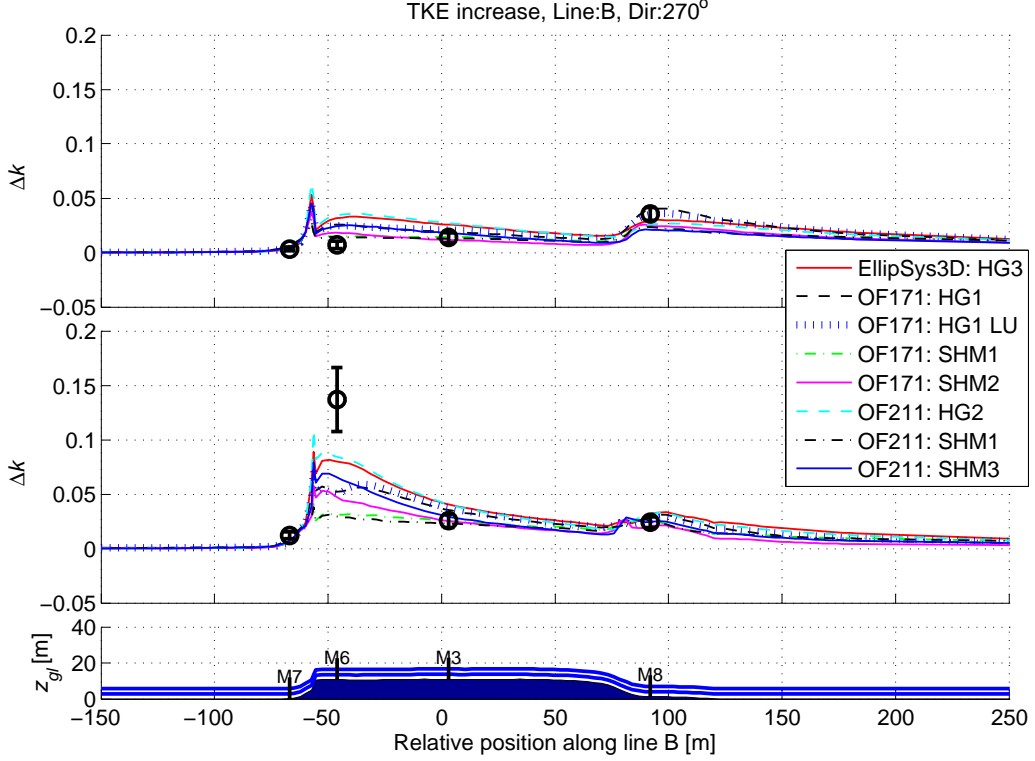

**Figure 10.** Bolund hill: Turbulent kinetic energy along the line B. Uppermost subfigure $5\,\mathrm{m}$ above the ground level, middle subfigure $2\,\mathrm{m}$ above the ground level.

`PCG` pressure solver also (results in parentheses in tab. 6) instead of `GAMG`. The grid sequencing procedure is analogous to the one presented in section 3.1.6, only here a grid corresponding to grid level 2 in EllipSys3D (every second point in all 3 directions is removed - the HG0 grid) has been separately created and utilized for the mesh sequencing procedure in the OpenFOAM HG(1-2) runs. The coarse SHM0 grid is created for the same purpose according to the specifications

5 in tab. 5 with regard to SnappyHexMesh based OpenFOAM simulations.

The obtained computational times are shown in tab. 6.

Comparing EllipSys3D to fastest OpenFOAM runs on grids of similar size (tab. 6), it can be observed that EllipSys3D is app. factor $4-6$ times faster, when grid sequencing procedure is turned on and factor $1.3-1.7$ times slower, when grid sequencing procedure is turned off, in obtaining the numerical solution of the same level of accuracy.

10 **4 Discussion**

First, it should be pointed out that several simulations using the Nikuradse's equivalent sand roughness length wall-model have been rerun in the more recent OpenFOAM v.2.1.1. Also, the OpenFOAM v.2.1.1 has been compiled using the




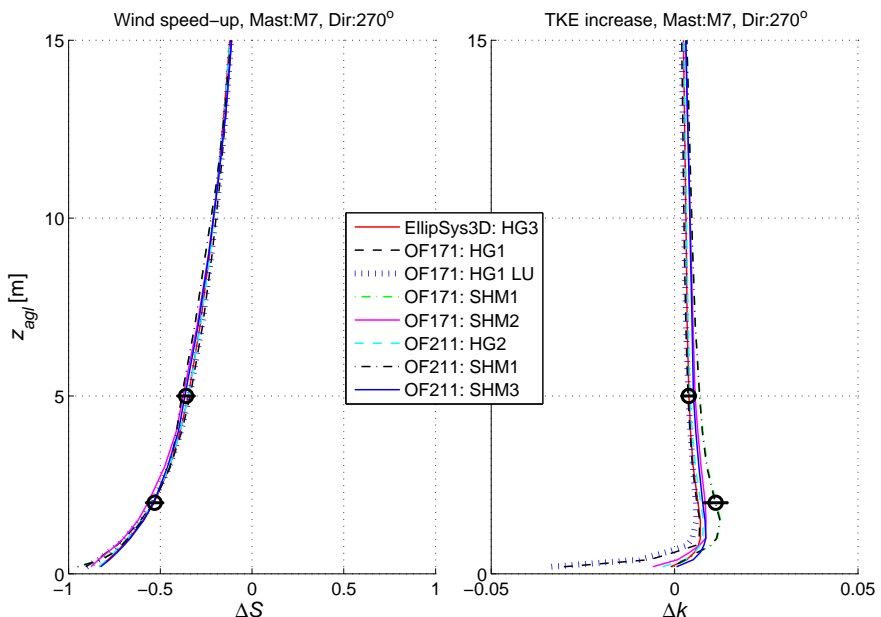

**Figure 11.** Bolund hill: Mast position 7, Speed-up and Turbulent kinetic energy

|  |  |  | Grid sequencing ON | Grid sequencing OFF | QUICK(V) | linear UpwindV | Grid Size (mill.) |
|---|---|---|---|---|---|---|---|
| EllipSys3D : | HG3 | $\Delta z = 0.03\,z_{0(\mathrm{L})}$ | 286 s | 3656 s | X |  | 6.29 |
| OF v.1.7.1 : | HG0 | $\Delta z = 25.0\,z_{0(\mathrm{L})}$ | - | 734 s | X |  | 0.78 |
| OF v.1.7.1 : | HG1 | $\Delta z = 12.5\,z_{0(\mathrm{L})}$ | 4073 s | 14396 s | X |  | 6.29 |
| OF v.1.7.1 : | HG1 | $\Delta z = 12.5\,z_{0(\mathrm{L})}$ | 3307 s | 9838 s |  | X | 6.29 |
| OF v.1.7.1 : | SHM0 | $\Delta z = 13.3\,z_{0(\mathrm{L})}$ | - | 119 s |  | X | 0.84 |
| OF v.1.7.1 : | SHM1 | $\Delta z = 10.0\,z_{0(\mathrm{L})}$ | 1844 s | 2108 s |  | X | 5.60 |
| OF v.1.7.1 : | SHM2 | $\Delta z = 13.3\,z_{0(\mathrm{L})}$ | 1507 s | 1700 s |  | X | 4.68 |
| OF v.2.1.1 : | HG2 | $\Delta z = 0.83\,z_{0(\mathrm{L})}$ | 10153 (5780) s | 40630 (33567) s | X |  | 6.29 |
| OF v.2.1.1 : | HG2 | $\Delta z = 0.83\,z_{0(\mathrm{L})}$ | 10259 (5650) s | 35857 (27683) s |  | X | 6.29 |
| OF v.2.1.1 : | SHM1 | $\Delta z = 10.0\,z_{0(\mathrm{L})}$ | 1171 s | 1789 s |  | X | 5.60 |
| OF v.2.1.1 : | SHM3 | $\Delta z = 1.00\,z_{0(\mathrm{L})}$ | 1656 s | 2820 s |  | X | 7.63 |

**Table 6.** Bolund hill: Simulation times for EllipSys3D and OpenFOAM codes. For information about different grids utilized in the computations see tab. 5 and fig. 8. The OpenFOAM v.2.2.1 runs on HG2 grid are conducted using the `PCG` pressure solver (results in parentheses) instead of `GAMG`.



optimized **Intel icc** compiler for the **Intel Xeon**$^{\circledR}$ (enclosed in the DTU Wind Energy cluster facility), CPU platform and selected simulations were rerun in this environment as well. In both cases no noticeable differences in the computational times, compared to the ones presented here, were observed.

## 4.1 Mesh generation

### 4.1.1 SnappyHexMesh

OpenFOAM's own SnappyHexMesh meshing tool seems to have reasonable grid generation applicability and flexibility for the investigated neutral atmospheric boundary layer (ABL) flow over complex terrain, although a number of problems were encountered during the meshing process.

In the Askervein hill case it was possible to create several grids with good general and boundary layer resolution capabilities using it. Dedicated grids for both Nikuradse's equivalent sand roughness length wall-model and the atmospheric roughness length wall-function approach could be made directly. Generating the surface layer, crucial for appropriate boundary layer description in the ABL simulations, was a quite difficult task. Only usable results were obtained by splitting the - *add surface layer* grid generation process form the rest of the SnappyHexMesh mesh generation procedure and disabling all mesh quality checks during this phase. Otherwise, very strange results with several regions of missing surface layer parts were obtained. The grids created during present work basically reflect more a limit in what SnappyHexMesh is capable of handling regarding the addition of surface layers in the grid, rather than a carefully considered user based specification of sizes and extent of different surface layer parameters.

In the Bolund hill case, some of the mentioned surface layer generation problems were even more emphasized. The abrupt change in surface structure at the Bolund hill front side created severe problems in the generation of the surface layers, so e.g. extending the `refinementSurfaces` level parameter to more than $5$, in order to better approximate the Bolund ground surface, was not possible in the current case. Basically, there is a very little freedom in specification of many surface layer related parameters. Generally, it seems very difficult as a user to be in control of the surface layer creation process using the SnappyHexMesh tool, making it quite difficult to be used consistently in relation to grid generation processes relevant for ABL flows.

### 4.1.2 HypGrid

In contrast HypGrid, a meshing tool developed deliberately to create low skewness hyperbolic 3D meshes based on complex surface topologies, appears to be able to cope with both investigated geometries without significant problems. EllipSys3D could easily handle all grids created by HypGrid, while some adjustments, described in section 3.2.2, were necessary in order to make suitable grids for OpenFOAM runs.



## 4.2 Accuracy

### 4.2.1 Askervein hill case

The results obtained with OpenFOAM for the Askervein hill case show a very good general agreement with EllipSys3D and cup anemometer measurements in terms of speed-up curves presented in fig. 4b). Especially results using Open-FOAM v.2.1.1 SHM(2-4) grids, OpenFOAM v.1.7.1 SHM(3-4) grids and EllipSys3D HG2 grid (fig. 4a)), seem to have a very good correspondence, both prior and after the hill top. The OpenFOAM (v.1.7.1 and v.2.1.1) calculation based on HypGrid created mesh (HG1) together with EllipSys3D calculation on the same grid appear to deviate significantly from the above mentioned cases and measurements on the lee side of the hill. Results based on OpenFOAM v.1.7.1 SHM1 grid seem to be placed in between the two above described sets of results. An important thing to note here is a very good correspondence between results of OpenFOAM v.2.1.1 and EllipSys3D, which are run on identical computational grids (HG1) using the same wall-function modeling approach[6].

Regarding the TKE plots presented on fig. 5, a good correspondence between all computations (and partially measurements) can be observed on the front side of the hill. Only ElipSys3D HG2 grid based calculation, deviate from the rest of the computations in the immediate vicinity of the hill top. A similar behavior regarding the TKE in this particular region has been observed in the OpenFOAM v.2.1.1. calculations on previously mentioned grids with cell center heights in order of the roughness length - $\Delta z = 0.83\,z_0$ and $\Delta z = 1.5\,z_0$ (not shown here).

The general deviations between numerical results on the lee side of the hill are much larger, but difference between all computations and measurements here appear much more significant (numerical findings under predict measurements by more than 50%) and dominant, than the differences between the computations. In the obtained steady state solutions occurrence of the flow separation has not been detected. As intermittent flow separation seemed to occur during the observational period and as separation increases the general turbulence levels, this can be a possible reason why currently used RANS turbulence model can not predict levels of turbulent kinetic energy on the lee side of the hill accurately (Undheim et al. (2006)). The RANS models in general are reported to have a substantial problem in predicting the measured turbulent kinetic energy levels correctly in the mentioned zone (Sørensen (1995); Kim and Patel (2000); Eidsvik (2005)). However, Castro et al. (2003) - using a high-order accurate schemes and Unsteady (U)RANS formulation, seem to better capture the measured turbulence properties whereas recent Large Eddy Simulation (LES) studies (Chow and Street (2009)) and Hybrid RANS/LES studies (Bechmann and Sorensen (2011)) do show some general, significant and promising improvements in this regard.

Considering OpenFOAM results from both fig. 4 and 5, the OpenFOAM computations, based on SnappyHexMesh created grids SHM(2-4), seem to have the best agreement with measurements and EllipSys3D.

---

[6]Practically only difference can be seen for horizontal distance from HT $> 500$ m.





### 4.2.2 Bolund hill case

The first observation regarding the Bolund hill results presented in figs. 9 and 10 is a very close agreement between results of OpenFOAM v.2.1.1 (HG2 grid, dashed cyan colored line) and EllipSys3D (HG3 grid, red line), which both use Richards and Hoxey (1993) based wall-function approach. This close agreement indicates indeed, that both flow solvers

are generally quite capable of producing reliable CFD results.

The difference in wall-modeling approach appear to play a dominant role in flow predictions in the recirculation zone on the lee side of the hill. Here, the OpenFOAM v.1.7.1 based calculations (HG1 (HG1 LU) and SHM2) seem to collapse to a single curve, while OpenFOAM v.2.1.1 based calculations (HG2, SHM(1,3) grids and EllipSys3D HG3 grid) seem to collapse to another curve at both $2$ and $5$ m above the ground level. Only the OpenFOAM v.1.7.1 SHM1 grid

based calculation follow the EllipSys3D and OpenFOAM v.2.1.1 results rather than the rest of the OpenFOAM v.1.7.1 computations.

Regarding the TKE plots in fig. 10 it is seen that at position of mast 6 (M6) all simulations seem to have a difficulty in producing the correct level of the turbulent kinetic energy especially for the measured height of $2$ m over the ground level. Considerable variation between the results can be seen in this particular region, where almost all OpenFOAM v.2.1.1 and

EllipSys3D results predict higher peak values close to M6 position than the OpenFOAM v.1.7.1 based calculations. The SHM1 grid based calculation (now in OpenFOAM v.2.1.1) deviates again from general tendencies and is seen to produce almost identical results as OpenFOAM v.1.7.1 calculation on the same grid.

The behavior of computed results on SHM1 grid appear therefore more to reflect the ability of the grid to "*snap*" the correct Bolund ground surface (`refinementSurface` level 3 is used in SHM1 grid vs. `refinementSurface` level

5 used in SHM(2,3) grids) rather than wall-modeling approach used in the computations. This inability of SHM1 grid to properly represent the Bolund ground surface is also visible in results at mast position M7 (fig. 11) especially on the TKE plot subfigure. It should be noted that increasing `refinementSurface` level to a value higher than 5 resulted in an inability of SnappyHexMesh to create any valid surface layers, due to grid distortion of closely spaced almost perpendicular cell layers, underlining a general difficulty/inability of full user control over surface grid layer creation in

SnappyHexMesh OpenFOAM utility. Therefore, the OpenFOAM v.2.1.1 SHM3 based results represent the highest peak value close to M6 position which can be produced with current SnappyHexMesh based setup. The OpenFOAM v.2.1.1 based HG2 computation is seen to easily extend the SHM3 mesh predicted peek value in this particular region.

Comparing OpenFOAM v.1.7.1 (HG1 and HG1 LU) based results which use identical settings and grid, with only difference in discretization scheme used (`QUICKV` vs. `linearUpwindV`) shows practically no difference in the obtained

results, indicating that utilization of `linearUpwindV` scheme for SnappyHexMesh based computations does not seem to impair quality of the computed results.



### 4.3  Speed

#### 4.3.1  Askervein hill case

Computational times from tab. 3 show that a converged solution on a SnappyHexMesh based grid SHM3 is between $3.5 - 6.5$ (Grid sequencing ON) and $5 - 7$ (Grid sequencing OFF) times faster to obtained than results on a HypGrid

based grid (HG1) with similar number of grid points, using both OpenFOAM v.1.7.1 and v.2.1.1 solvers. A closer look on the residual curves showed that a single iteration takes roughly the same amount of time in both SnappyHexMesh and HypGrid based cases, but the number of iterations needed to reach the convergence level of $2\,\mathrm{x}\,10^{-4}$ is, as indicated in tab. 3, much higher in the HypGrid based mesh case. This indicates that a structured meshing tool like HypGrid might not be the most optimal choice for grid generation purposes in OpenFOAM.

Focusing now on SnappyHexMesh based results, it can be seen that speed increase close to factor $2$ is gained by using the grid sequencing procedure. The same is almost true in the HypGrid based cases - slightly lower speed gain in the OpenFOAM v.1.7.1 case and slightly higher in the OpenFOAM v.2.1.1 case. In the SnappyHexMesh SHM3 case the speed differences between OpenFOAM v.2.1.1 and OpenFOAM v.1.7.1 computations are almost negligible, indicating that differences in the wall modeling approach used in the two calculations does not seem to effect the convergence

speed process significantly. Also considering the increase in the number of grid points - following cases SHM(1,3,4) OpenFOAM v.1.7.1 and SHM(2,3,4) OpenFOAM v.2.1.1 a proportional reflection in the computational times from tab. 3 can be observed, although the proportionality factor seem to vary in a non linear manner.

Compared to the Bolund hill test case, the grid sequencing procedure in the EllipSys3D computations does not appear to influence the computational time considerably. The reasons for this probably lie in the general flow complexity (or rather

lack of the same) of the Askervein hill case, as it apparently prevents the solver to fully utilize the advantage of a solution on the coarser grid level, compared to using the standard start guess in the solution procedure on the fine grid. Placing the first near wall cell very close to the ground - HG2 grid, is seen to have a slight influence on the computational times also (relative to HG1 grid).

Comparing the EllipSys3D and OpenFOAM runs, best done on a grids of similar size and position of the first cell center

above the ground level - the HG1 and SHM3 grids, it is seen that EllipSys3D is app. $1.9$ (Grid sequencing ON) and $2.5$ (Grid sequencing OFF) times faster in obtaining the converged solution.

#### 4.3.2  Bolund hill case

In the Bolund hill case[7], a significant difference in computational times between SnappyHexMesh and HypGrid based OpenFOAM cases is again observed - tab. 6. Comparing results on grids of similar sizes and near ground surface

resolution capabilities, conducted using the same `linearUpwindV` discretization scheme, a speed difference factor of $1.8$ (Grid sequencing ON), $4.7$ (Grid sequencing OFF) - OpenFOAM 1.7.1 (cases HG1 and SHM1) and $3.4$ (Grid sequencing ON), $9.8$ (Grid sequencing OFF) - OpenFOAM 2.1.1 (cases HG2 - fastest, `PCG` based solution and SHM3)

---

[7]A residual convergence level of $10^{-4}$ is reached in all computations.





can be observed. This underlines once again a general OpenFOAM issue with grids comprising of high aspect-ratio cells. It is also seen from tab. 6 that multigrid `GAMG` pressure solver is not functioning optimally and is significantly outperformed by the more conventional `PCG` solver in HypGrid based OpenFOAM v.2.1.1 cases.

Furthermore, tab. 6 indicates that converged solution using `QUICKV` scheme is slower to obtain than the corresponding
solution using `linearUpwindV` discretization scheme, especially when grid sequencing procedure is turned OFF (in order of $20-30\%$).

Grid sequencing procedure seem to have a smaller positive influence on the SnappyHexMesh based calculations in the Bolund hill case (speed-up factor $< 2$), compared to the Askervein hill case, while the opposite is true for the HypGrid based calculations (speed-up factor app. $3-5$).

The EllipSys3D code seem to benefit considerably from its automatic grid sequencing procedure, as it speeds up the computations by more than a factor of $10$.

Comparing the EllipSys3D run with best directly comparable OpenFOAM run (v.2.1.1 SHM3 smallest $\Delta z$, identical wall-modeling approach) it can be seen that EllipSys3D is app. factor $6$ times faster in obtaining the converged solution (Grid sequencing ON), but despite the fact that OpenFOAM SHM3 computation includes almost $20\%$ more grid points, it
is faster (app. $30\%$) to obtain the solution on it than using the structured EllipSys3D solver with grid sequencing turned OFF.

## 5  Conclusion

In this work, the unstructured OpenFOAM flow solver is compared to the structured EllipSys3D flow solver on two test cases calculating neutral atmospheric boundary layer flow over complex terrain.

Two meshing tools are considered, the structured hyperbolic 3D mesh generator HypGrid and OpenFOAM's own hexahedral mesh generator SnappyHexMesh. OpenFOAM was found to be able to successfully perform calculations on the HypGrid created meshes in both considered test cases. The SnappyHexMesh could also produce reasonable grids in both Askervein hill and Bolund hill test cases. A very important parameter for computational grids in ABL flows - height of the first near ground cell, proved to be very difficult to directly control using the SnappyHexMesh tool reflected in its
(in)ability to create surface layers. This issue makes it quite difficult to use SnappyHexMesh consistently in relation to grid generation processes relevant for ABL flows, so a tool with more direct user control over this crucial part of the grid generation process can be recommended.

In terms of accuracy, both flow solvers are found to perform equally well on the two test cases, both regarding the mean flow velocity and turbulence quantities. Especially OpenFOAM v.2.1.1 and EllipSys3D calculations, performed on identical
(Askervein hill case) and very similar (Bolund hill case) computational grids, using the same approach to wall-function modeling of ABL flows (Richards and Hoxey (1993)) were found to have a great mutual correspondence, underlying that both flow solvers are quite capable of producing reliable numerical results.



However, a large discrepancy in the speed performance is found. A very large difference in calculation times is obtained between HypGrid and SnappyHexMesh based OpenFOAM calculations, indicating that a structured meshing tools, typically creating grids with high aspect ratio cells in ABL flows, causes OpenFOAM solver to perform inefficiently. Results of the present work show that this performance issue can be partially addressed by introducing a grid sequencing procedure

in the OpenFOAM runs. Generally, the grid sequencing procedure had a very positive effect on almost all OpenFOAM computations and can be highly recommended.

Comparing EllipSys3D, which utilizes the grid sequencing procedure by default, and OpenFOAM SnappyHexMesh based calculation times, the structured EllipSys3D solver is found to perform app. $2-6$ times faster on grids with similar properties.

Generally, the overall OpenFOAM performance is found to be quite good. Using a combination of hexahedral and polyhedral cells the SnappyHexMesh tool can produce suitable grids in many relevant ABL flow cases and bring the computational times close to the level of the structured EllipSys3D code, but inability to fully control ground surface approximation and accompanied surface layer creation, as in the Bolund hill case, can potentially impair the quality of the obtained SnappyHexMesh based results. On the other hand, using the structured HypGrid solver, where terrain

description and surface layer creation can be fully controlled, proved to be accompanied by a high speed performance penalty in the OpenFOAM runs. Thus, an unstructured mesh generation tool where both surface approximation and accompanied surface layer creation is better controlled might be an optimal meshing tool for ABL flow calculations using the OpenFOAM solver.

*Acknowledgements.* This work is supported by the Center for Computational Wind Turbine Aerodynamics and Atmospheric Turbulence

funded by the Danish Council for Strategic Research, grant number 09-067216. Partial support from the ERANET Plus proejct called the New European Wind Atlas is appreciated.





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
