# Peer review of "Comparison of OpenFOAM and EllipSys3D for neutral atmospheric flow over complex terrain"

_Wind Energy Science, 2016_

## Referee Comment (RC1) · Anonymous Referee #1 · 22 Mar 2016

General comments :

This article presents a systematic comparison (almost the same mathematical model, same number of points, etc.) between EllipSys3D and OpenFOAM. The article is of good quality, the results presented are interesting since they are related to 2 important computer codes typically used in the wind energy scientific and industrial communities. The topic discussed (effort to produce appropriate grid, effort of producing results, accuracy of results, computer time) is particularly of high importance to the wind energy industrial community.

I believe it is worth publishing in an archive journal once the following minor comments/errors are treated properly:

p.3, line 4: epsilon is the dissipation rate, not dissipation. Please make appropriate changes elsewhere in the text.

p.3, line 24, equation 4: why are you using s here? This is a velocity, but it might be confused with the speedup you are presenting on Fig. 4.

p.4, line 5: I think you should remove the word 'kinematic'.

p.4, line 6, equation (6): adding a figure may facilitate locating the position of the point 'o'.

p.4, line 11, equation (8): is it the expression for epsilon at point 'o' or at the bottom face at $z\_0$? Please make it clear in the final version.

p.4, lines 12-13: you are mentioning 'a von Neumann boundary condition'. This is incomplete information. What is the value of the gradient applied there?

p.5, line 17: equation (11) is different than (8). Is it a typo? Please clarify by commenting in the final version. It is not clear to me since you are writing here that the mathermatical models are identical. Boundary conditions are part of the model for me. Similar comment for Eqs. (4) - (9) and Eqs. (10) - (12).

p.10, Table 2: Is Delta_z the same as in equation (7)?

p.21, line 13: There is a typo here. You have written ElipSys3D.

p. 23, line 24: remove 'a' in front of 'grids'.

---

## Referee Comment (RC2) · Anonymous Referee #2 · 25 Mar 2016

The main subject of the article is the comparison between two different CFD codes for simulating neutral atmospheric conditions over a complex terrain using RANS equations (steady state and k-epsilon turbulence model). The codes (EllipSys3D and OpenFOAM) are compared on the basis of mesh generation (time spent, quality and "easyness" to use), resolution time and results. Two test cases are discussed, the Askervein hill and the Bolund island.

The paper is well written, of good quality and giving interesting information for CFD modelers in the wind energy sector. Authors made the choice to discuss CFD methods without describing the flow behavior nor comparing to state-of-art simulations (not even cited). Therefore, I wonder if it is in the scope of the journal. A CFD method oriented

journal may be more suited.

General comments:

1- The current version of OpenFOAM is OF-3.0.1. Is the version OF-1.7.1 (released in 26th August 2010) still an academic/industrial standard? In my opinion, the discussion about this old OF version may be removed as it does not reflect the standard use of OF. The comparison is also bias by the fact that this given version doesn't include Richard and Hoxey wall function. The other version used OF-2.1.1 (from may 2012), is more recent but does it reflect the "state-of-art" of OF developments? Would OF-3 include new functions for mesh generation? What is the version (year) of EllipSys3D?

2- Numerous function are mentioned and it is sometimes difficult to know from which solver they belong. I would suggest to make a simple table defining the function of each solver.

3- Several fonts/format are used to underline function/solver names such as surfgrid, blockMesh... Please check carefully the consistency in the text (ie p 7 and 8 "HypGrid" and HypGrid are mentionned.) Why snappyHexMesh and blockMesh are not in the same font?

4- It has to be noted that the work is authored by some of the developers of EllipSys3D, the comparison of the two codes is therefore made between a developer for EllipSys and by a user for OpenFOAM.

5- The comparison is sometimes not very precise "was quite a quite difficult task" (p20, l12 ), "seems very difficult as a user..." (p.20, l.22), and it is unclear whether is come from a code limitation of less experience with the meshing procedure. Comparison on more objective criteria is desirable.

Specific comments:

Figures 1, 2, 3, and 6, 7, 8. The visualization of the OF grid generated is surprising, I believe the authors made an appropriated grid but the figure looks weird. It looks like

the visualization is not appropriately done (triangulated by Paraview?). Additionally, the view chosen for the OF grid figures tend to diminish its quality. Please use the same kind of figure for both: view of the surface (like fig 1) and one vertical plane.

p3 eq(4), it is quite surprising to use 's' for mean wind-speed. U may be more appropriated in this context. Please change everywhere in the text.

Table 2: Could you consider a better way to include columns titles.

Fig 4 and 5. It would be interesting to compare the "best" EllipSys3D (HG2?) with the "best" OF(211-SHM4?). Same thing for Bolund.

---

## Author Comment (AC1) · 4 Apr 2016

Thank you for the comments.

We will below give our comments/response to each of the outlined points/issues.

- p.3, line 4 − dissipation is changed to dissipation rate, also elsewhere in the text.

- p.3, line 24 − s is changed to $\mathbf{u}$, also elsewhere in the text.

- p.4, line 5 − kinematic is removed from the text.

- p.4, lines 6 and 11:

  o The applied modelling strategy leads to presented equations (4) and (5) on p.3 in the text. From each of the equations the $u_*$ can be deduced. $u_*$ deduced from equation (5) is called $u_0$, and $u_*$ deduced form equation (4) is called $u_w$. They are both used in determining the wall shear stress, as indicated on p.4, line 5. Therefore, both $u_0$ and $u_w$ are representing the same physical quantity. In order to clarify and emphasize the above mentioned, we are trying to use different notation/symbols here. If this is still unclear, will using $u_{w_1}$ and $u_{w_2}$ instead of $u_0$ and $u_w$ help?

- p.4, lines 12-13 − formulation: 'von Neumann' is changed to zero gradient.

- p.5, line 17:

  o Equation (11) refers to Nikuradse's equivalent sand roughness model. The difference between this modelling approach and Richards and Hoxey model used in OpenFoam 2.1.1 and EllipSys3D is commented on p. 4, line 23:
  "Comparing with (Eq. 4) it is seen that the roughness for OpenFOAM v.1.7.1 is placed on top of the wall ($\mathbf{u} = 0$ for $z = z_0$)",
  and that is the reason why $z_0$ is missing from equation (11) compared to equation (8), and correspondingly leads to differences between equations (4) − (9) and (10) − (12) regarding the $z_0$.

- p.10, Table 2 − all references to $\Delta z$ in the article refer to the first cell center height above the wall boundary.

- p. 21, line 13 − Typo corrected.

- p. 23, line 24 − 'a' removed

[revised manuscript text omitted]

20    ogy*, 120(3):477–495, 2006. ISSN 15731472, 00068314. doi:10.1007/s10546-006-9065-5.

---

## Author Comment (AC2) · 4 Apr 2016

Thank you for the comments.

We will below give our comments/response to each of the outlined points/issues.

General comments:

1) As indicated on p. 19 lines 18-22 (in the revised article version), we have rerun (basically all) OpenFOAM 1.7.1 cases using Nikuradse's equivalent sand roughness model in OpenFOAM 2.1.1, without noticeable differences in the computational times – compared to ones presented for OF-1.7.1 in the article. Retaining the reference to two different OF versions is basically only meant for emphasizing the difference in the wall-functions approaches utilized in the comparisons.
It is true that a newer OF version (OF 3.0.1) is available now, but given the fact that OF computational grids still have a restriction on the cell aspect ratio, which is a quite important and dominating factor regarding the whole OF based computation chain process with respect to ABL relevant flows, we are quite certain that the presented findings still reflect the current state of the OpenFOAM's applicability for ABL relevant flow cases. Also given the fact that the OF source code core is significantly changing for almost each new version/release, and considerable effort is typically needed to move/port all user defined routines/cases/code changes to a new OF version, it is our understanding, that many "old" OpenFOAM versions are still used among modelers in the wind energy community.
Regarding the mesh generation, comparing the current SnappyHexMesh capabilities to the ones used in our work, we do see that some improvements have been made to the Layer Addition (OF 2.4.0) process that can be relevant for the present work, but given the complexity of the whole grid generation process, we do not think that improvement of single feature would change/significantly impact findings presented on p. 19-20 (in the revised article version).
The EllipSys3D version used in this work is also from 2012.

2) Surfgrid and HypGrid are the only non-OpenFOAM CFD tools mentioned in the article text. We removed the (verb) emphasis on them throughout the text, and hope that this can resolve the mentioned issue. If still necessary, the table can be added to the main article text.

3) We have now revised the article and only emphasis on OF tools/utilities/functions are left underlined. If this is more confusing than improving the article readability, this emphasis can be easily removed.

4) Completely true. For that reason, we have tried to present as much information as feasible in the article text, so as many as possible OF parameters/functions/tools used, could be identified. We have tried to follow the available best OF practices, but if some more advanced OF users or developers can suggest changes/improvements in this regard, we'll be happy if our work can start this type of a discussion, probably relevant for many modelers in the wind energy community.

5) As already emphasized under comment 4., the authors are only users of the OpenFOAM CFD toolbox and can in principle make mentioned comments from this - user perspective.

Specific comments:

- A volume grid and surface grid views for snappyHexMesh created computational meshes are added to the article text in Figures 1 and 6. Indicators relevant for understanding many snappyHexMesh parameters presented in Tables 2 and 5 were very difficult to include in those figures, so Paraview based Figures 3 and 8 are retained in the text.

- s is changed to $\boldsymbol{u}$, also elsewhere in the text.

- Table 2 and 5 now have a more appropriate column titles.

- OF 2.1.1 SHM4 results are added to Figures 4a) and 5a). Note that the good general agreement between OF and EllipSys results is already commented on p. 20 lines 21-24 (in the revised article version). All Bolund – both OF and EllipSys3D cases are already included in Figures 9 and 10.

[revised manuscript text omitted]

20   ogy*, 120(3):477–495, 2006. ISSN 15731472, 00068314. doi:10.1007/s10546-006-9065-5.

---

## Author Response (AR1)

Dear Associated and Chef Editor,

Thank you for the comments regarding final improvements of our manuscript.

All mentioned issues are addressed and integrated into the latest manuscript version.

Best Regards

Dalibor Cavar